# Graph Matching via Multiplicative Update Algorithm

**Bo Jiang**
School of Computer Science
and Technology
Anhui University, China
jiangbo@ahu.edu.cn

**Jin Tang**
School of Computer Science
and Technology
Anhui University, China
tj@ahu.edu.cn

**Chris Ding**
CSE Department,
University of Texas at
Arlington, Arlington, USA
chqding@uta.edu

**Yihong Gong**
School of Electronic
and Information Engineering
Xi'an Jiaotong University, China
ygong@mail.xjtu.edu.cn

**Bin Luo**
School of Computer Science
and Technology,
Anhui University, China
luobin@ahu.edu.cn

## Abstract

As a fundamental problem in computer vision, graph matching problem can usually be formulated as a Quadratic Programming (QP) problem with doubly stochastic and discrete (integer) constraints. Since it is NP-hard, approximate algorithms are required. In this paper, we present a new algorithm, called Multiplicative Update Graph Matching (MPGM), that develops a multiplicative update technique to solve the QP matching problem. MPGM has three main benefits: (1) theoretically, MPGM solves the general QP problem with doubly stochastic constraint naturally whose convergence and KKT optimality are guaranteed. (2) Empirically, MPGM generally returns a sparse solution and thus can also incorporate the discrete constraint approximately. (3) It is efficient and simple to implement. Experimental results show the benefits of MPGM algorithm.

## 1 Introduction

In computer vision and machine learning area, many problems of interest can be formulated by graph matching problem. Previous approaches [3–5, 15, 16] have formulated graph matching as a Quadratic Programming (QP) problem with both doubly stochastic and discrete constraints. Since it is known to be NP-hard, many approximate algorithms have been developed to find approximate solutions for this problem [8, 16, 21, 24, 20, 13].

One kind of approximate methods generally first develop a continuous problem by relaxing the discrete constraint and aim to find the optimal solution for this continuous problem. After that, they obtain the final discrete solution by using a discretization step such as Hungarian or greedy algorithm [3, 15, 16]. Obviously, the discretization step of these methods is generally independent of the matching objective optimization process which may lead to weak local optimum for the problem. Another kind of methods aim to obtain a discrete solution for QP matching problem [16, 1, 24]. For example, Leordeanu et al. [16] proposed an iterative matching method (IPFP) which optimized the QP matching problem in a discrete domain. Zhou et al. [24, 25] proposed an effective graph matching method (FGM) which optimized the QP matching problem approximately using a convex-concave relaxation technique [21] and thus returns a discrete solution for the problem. From optimization aspect, the core optimization algorithm used in both IPFP [16] and FGM [24] is related to Frank-Wolfe [9] algorithm and FGM [24, 25] further uses a path following procedure to alleviate the local-optimum problem more carefully. The core of Frank-Wolfe [9] algorithm is to optimize the quadratic problem by sequentially optimizing the linear approximations of QP problem. In addition

to optimization-based methods, probabilistic methods can also be used for solving graph matching problems [3, 19, 23].

In this paper, we propose a new algorithm, called Multiplicative Update Graph Matching (MPGM), that develops a multiplicative update technique for the general QP problem with doubly stochastic constraint. Generally, MPGM has the following three main aspects. First, MPGM solves the general QP problem with doubly stochastic constraint directly and naturally. In MPGM algorithm, each update step has a closed-form solution and the convergence of the algorithm is also guaranteed. Moreover, the converged solution is guaranteed to be Karush-Kuhn-Tucker (KKT) optimality. Second, empirically, MPGM can generate a sparse solution and thus incorporates the discrete constraint naturally in optimization. Therefore, MPGM can obtain a local optimal discrete solution for the QP matching problem. Third, it is efficient and simple to implement. Experimental results on both synthetic and real-world matching tasks demonstrate the effectiveness and benefits of the proposed MPGM algorithm.

## 2    Problem Formulation and Related Works

**Problem Formulation.** Assume $G = (V, E)$ and $G' = (V', E')$ are two attributed graphs to be matched, where each node $v_i \in V$ or edge $e_{ik} \in E$ has an attribute vector $\mathbf{a}_i$ or $\mathbf{r}_{ik}$. The aim of graph matching problem is to establish the correct correspondences between $V$ and $V'$. For each correspondence $(v_i, v'_j)$, there is an affinity $S_a(\mathbf{a}_i, \mathbf{a}'_j)$ that measures how well node $v_i \in V$ matches node $v'_j \in V'$. Also, for each correspondence pair $(v_i, v'_j)$ and $(v_k, v'_l)$, there is an affinity $S_r(\mathbf{r}_{ik}, \mathbf{r}'_{jl})$ that measures the compatibility between node pair $(v_i, v_k)$ and $(v'_j, v'_l)$. One can define an affinity matrix $\mathbf{W}$ whose diagonal term $\mathbf{W}_{ij,ij}$ represents $S_a(\mathbf{a}_i, \mathbf{a}'_j)$, and the non-diagonal element $\mathbf{W}_{ij,kl}$ contains $S_r(\mathbf{r}_{ik}, \mathbf{r}'_{jl})$. The one-to-one correspondences can be represented by a permutation matrix $\mathbf{X} \in \{0, 1\}^{n \times n}$, where $n = |V| = |V'|$[1]. Here, $\mathbf{X}_{ij} = 1$ implies that node $v_i$ in $G$ corresponds to node $v'_j$ in $G'$, and $\mathbf{X}_{ij} = 0$ otherwise. In this paper, we denote $\mathbf{x} = (\mathbf{X}_{11}...\mathbf{X}_{n1}, ..., \mathbf{X}_{1n}...\mathbf{X}_{nn})^{\mathrm{T}}$ as a column-wise vectorized replica of $\mathbf{X}$. The graph matching problem is generally formulated as a Quadratic Programming (QP) problem with doubly stochastic and discrete constraints [16, 3, 10], i.e.,

$$\mathbf{x}^* = \arg\max_{\mathbf{x}} (\mathbf{x}^{\mathrm{T}} \mathbf{W} \mathbf{x}) \quad s.t. \ \mathbf{x} \in \mathcal{P}, \tag{1}$$

where $\mathcal{P}$ is defined as,

$$\mathcal{P} = \{\mathbf{x} \mid \forall i \ \textstyle\sum_{j=1}^n \mathbf{x}_{ij} = 1, \forall j \ \textstyle\sum_{i=1}^n \mathbf{x}_{ij} = 1, \mathbf{x}_{ij} \in \{0, 1\}\} \tag{2}$$

The above QP problem is NP-hard and thus approximate relaxations are usually required. One popular way is to relax the permutation domain $\mathcal{P}$ to the doubly stochastic domain $\mathcal{D}$,

$$\mathcal{D} = \{\mathbf{x} | \forall i \ \textstyle\sum_{j=1}^n \mathbf{x}_{ij} = 1, \forall j \ \textstyle\sum_{i=1}^n \mathbf{x}_{ij} = 1, \mathbf{x}_{ij} \geq 0\}. \tag{3}$$

That is solving the following relaxed matching problem [21, 20, 10],

$$\mathbf{x}^* = \arg\max_{\mathbf{x}} (\mathbf{x}^{\mathrm{T}} \mathbf{W} \mathbf{x}) \quad s.t. \ \mathbf{x} \in \mathcal{D}. \tag{4}$$

Since $\mathbf{W}$ is not necessarliy positive (or negative) semi-definite, thus this problem is generally not a concave or convex problem.

**Related Works.** Many algorithms have been proposed to find a local optimal solution for the above QP matching problem (Eq.(4)). One kind of popular methods is to use constraint relaxation and projection, such as GA [10] and RRWM [3]. Generally, they iteratively conduct the following two steps: (a) searching for a solution by ignoring the doubly stochastic constraint temporarily; (b) Projecting the current solution onto the desired doubly stochastic domain to obtain a feasible solution. Note that the projection step (b) is generally independent of the optimization step (a) and thus may lead to weak local optimum. Another kind of important methods is to use objective function approximation and thus solves the problem approximately, such as Frank-Wolfe algorithm [9]. Frank-Wolfe aims to optimize the above quadratic problem by sequentially solving the approximate linear problems. This algorithm has been widely adopted in many recent matching methods [16, 24, 21], such as IPFP [16] and FGM [24].

## 3 Algorithm

Our aim in this paper is to develop a new algorithm to solve the general QP matching problem Eq.(4). We call it as Multiplicative Update Graph Matching (MPGM). Formally, starting with an initial solution vector $\mathbf{x}^{(0)}$, MPGM solves the problem Eq.(4) by iteratively updating a current solution vector $\mathbf{x}^{(t)}, t = 0, 1...$ as follows,

$$\mathbf{x}_{kl}^{(t+1)} = \mathbf{x}_{kl}^{(t)} \left[ \frac{2(\mathbf{W}\mathbf{x}^{(t)})_{kl} + \mathbf{\Lambda}_k^- + \mathbf{\Gamma}_l^-}{\mathbf{\Lambda}_k^+ + \mathbf{\Gamma}_l^+} \right]^{1/2}, \tag{5}$$

where $\mathbf{\Lambda}_k^+ = (|\mathbf{\Lambda}_k| + \mathbf{\Lambda}_k)/2, \mathbf{\Lambda}_k^- = (|\mathbf{\Lambda}_k| - \mathbf{\Lambda}_k)/2, \mathbf{\Gamma}_k^+ = (|\mathbf{\Gamma}_k| + \mathbf{\Gamma}_k)/2, \mathbf{\Gamma}_k^- = (|\mathbf{\Gamma}_k| - \mathbf{\Gamma}_k)/2$, and the Lagrangian multipliers $(\mathbf{\Lambda}, \mathbf{\Gamma})$ are computed as,

$$\mathbf{\Gamma} = 2\left(\mathbf{I} - \mathbf{X}^{(t)\mathrm{T}}\mathbf{X}^{(t)}\right)^{-1} \left[ \mathrm{diag}\left(\mathbf{K}^{(t)\mathrm{T}}\mathbf{X}^{(t)}\right) - \mathbf{X}^{(t)\mathrm{T}} \mathrm{diag}\left(\mathbf{K}^{(t)}\mathbf{X}^{(t)\mathrm{T}}\right) \right]$$

$$\mathbf{\Lambda} = 2 \mathrm{diag}\left(\mathbf{K}^{(t)}\mathbf{X}^{(t)\mathrm{T}}\right) - \mathbf{X}^{(t)}\mathbf{\Gamma} \tag{6}$$

where $\mathbf{K}^{(t)}, \mathbf{X}^{(t)}$ are the matrix forms of vector $(\mathbf{W}\mathbf{x}^{(t)})$ and $\mathbf{x}^{(t)}$, respectively, i.e., $\mathbf{K}^{(t)}, \mathbf{X}^{(t)} \in \mathbb{R}^{n \times n}$ and $\mathbf{K}_{kl}^{(t)} = (\mathbf{W}\mathbf{x}^{(t)})_{kl}, \mathbf{X}_{kl}^{(t)} = \mathbf{x}_{kl}^{(t)}$. $\mathbf{\Lambda} = (\mathbf{\Lambda}_1, \cdots \mathbf{\Lambda}_n)^{\mathrm{T}} \in \mathbb{R}^{n \times 1}, \mathbf{\Gamma} = (\mathbf{\Gamma}_1, \cdots \mathbf{\Gamma}_n)^{\mathrm{T}} \in \mathbb{R}^{n \times 1}$. The iteration starts with an initial $\mathbf{x}^{(0)}$ and is repeated until convergence.

**Complexity.** The main complexity in each iteration is on computing $\mathbf{W}\mathbf{x}^{(t)}$. Thus, the total computational complexity for MPGM is less than $O(MN^2)$, where $N = n^2$ is the length of vector $\mathbf{x}^{(t)}$ and $M$ is the maximum iteration. Our experience is that the algorithm converges quickly and the average maximum iteration $M$ is generally less than 200. Theoretically, the complexity of MPGM is the same with RRWM [3] and IPFP [16], but obviously lower than GA [10] and FGM [24].

**Comparison with Related Works.** Multiplicative update algorithms have been studied in solving matching problems [6, 13, 11, 12]. Our work is significantly different from previous works in the following aspects. Previous works [6, 13, 11] generally first develop a kind of approximation (or relaxation) for QP matching problem by ignoring the doubly stochastic constraint, and then aim to find the optimum of the relaxation problem by developing an algorithm. In contrast, our work focus on the general and challengeable QP problem with doubly stochastic constraint (Eq.(4)), and derive a simple multiplicative algorithm to solve the problem Eq.(4) directly. Note that, the proposed algorithm is not limited to solving QP matching problem only. It can also be used in some other QP (or general continuous objective function) problems with doubly stochastic constraint (e.g. MAP inference, clustering) in machine learning area. In this paper, we focus on graph matching problem.

**Starting Point.** To alleviate the local optima and provide a feasible starting point for MPGM algorithm, given an initial vector $\mathbf{x}^{(0)}$, we first use the simple projection $\mathbf{x}^{(0)} = P(\mathbf{W}\mathbf{x}^{(0)})$ several times to obtain a kind of the feasible start point for MPGM algorithm. Here $P$ denotes the projection [22] or normalization [20] to make $\mathbf{x}^{(0)}$ satisfy the doubly stochastic constraint.

## 4 Theoretical Analysis

**Theorem 1**. *Under update Eq.(5), the Lagrangian function $\mathcal{L}(\mathbf{x})$ is monotonically increasing,*

$$\mathcal{L}(\mathbf{x}) = \mathbf{x}^{\mathrm{T}}\mathbf{W}\mathbf{x} - \sum_{i=1}^{n} \mathbf{\Lambda}_i (\sum_{j=1}^{n} \mathbf{x}_{ij} - 1) - \sum_{j=1}^{n} \mathbf{\Gamma}_j (\sum_{i=1}^{n} \mathbf{x}_{ij} - 1) \tag{7}$$

*where $\mathbf{\Lambda}, \mathbf{\Gamma}$ are Lagrangian multipliers.*

**Proof**. To prove it, we use the auxiliary function approach [7, 14]. An auxiliary function function $\Phi(\mathbf{x}, \tilde{\mathbf{x}})$ of Lagrangian function $\mathcal{L}(\mathbf{x})$ satisfies following,

$$\Phi(\mathbf{x}, \mathbf{x}) = \mathcal{L}(\mathbf{x}), \Phi(\mathbf{x}, \tilde{\mathbf{x}}) \leq \mathcal{L}(\mathbf{x}). \tag{8}$$

Using the auxiliary function $\Phi(\mathbf{x}, \tilde{\mathbf{x}})$, we define

$$\mathbf{x}^{(t+1)} = \arg\max_{\mathbf{x}} \Phi(\mathbf{x}, \mathbf{x}^{(t)}). \tag{9}$$

Then by construction of $\Phi(\mathbf{x}, \tilde{\mathbf{x}})$, we have

$$\mathcal{L}(\mathbf{x}^{(t)}) = \Phi(\mathbf{x}^{(t)}, \mathbf{x}^{(t)}) \leq \mathcal{L}(\mathbf{x}^{(t+1)}). \tag{10}$$

This proves that $\mathcal{L}(\mathbf{x}^{(t)})$ is monotonically increasing.

The main step in the following of the proof is to provide an appropriate auxiliary function and find the global maximum for the auxiliary function. We rewrite Eq.(7) as

$$\mathcal{L}(\mathbf{x}) = \mathbf{x}^T \mathbf{W} \mathbf{x} - \sum_{i=1}^{n} \mathbf{\Lambda}_i (\sum_{j=1}^{n} \mathbf{x}_{ij} - 1) - \sum_{j=1}^{n} \mathbf{\Gamma}_j (\sum_{i=1}^{n} \mathbf{x}_{ij} - 1)$$

$$= \sum_{i=1}^{n} \sum_{j=1}^{n} \sum_{k=1}^{n} \sum_{l=1}^{n} \mathbf{W}_{ij,kl} \mathbf{x}_{ij} \mathbf{x}_{kl} - \sum_{i=1}^{n} \mathbf{\Lambda}_i (\sum_{j=1}^{n} \mathbf{x}_{ij} - 1) - \sum_{j=1}^{n} \mathbf{\Gamma}_j (\sum_{i=1}^{n} \mathbf{x}_{ij} - 1). \tag{11}$$

We show that one auxiliary function $\Phi(\mathbf{x}, \tilde{\mathbf{x}})$ of $\mathcal{L}(\mathbf{x})$ is,

$$\Phi(\mathbf{x}, \tilde{\mathbf{x}}) = \sum_{i=1}^{n} \sum_{j=1}^{n} \sum_{k=1}^{n} \sum_{l=1}^{n} \mathbf{W}_{ij,kl} \tilde{\mathbf{x}}_{ij} \tilde{\mathbf{x}}_{kl} \left( 1 + \log \frac{\mathbf{x}_{ij} \mathbf{x}_{kl}}{\tilde{\mathbf{x}}_{ij} \tilde{\mathbf{x}}_{kl}} \right) \tag{12}$$

$$- \sum_{i=1}^{n} \mathbf{\Lambda}_i^{+} \Big[ \sum_{j=1}^{n} \frac{1}{2} \big( \frac{\mathbf{x}_{ij}^2}{\tilde{\mathbf{x}}_{ij}} + \tilde{\mathbf{x}}_{ij} \big) - 1 \Big] + \sum_{i=1}^{n} \mathbf{\Lambda}_i^{-} \Big[ \sum_{j=1}^{n} \tilde{\mathbf{x}}_{ij} (1 + \log \frac{\mathbf{x}_{ij}}{\tilde{\mathbf{x}}_{ij}}) - 1 \Big]$$

$$- \sum_{j=1}^{n} \mathbf{\Gamma}_j^{+} \Big[ \sum_{i=1}^{n} \frac{1}{2} \big( \frac{\mathbf{x}_{ij}^2}{\tilde{\mathbf{x}}_{ij}} + \tilde{\mathbf{x}}_{ij} \big) - 1 \Big] + \sum_{j=1}^{n} \mathbf{\Gamma}_j^{-} \Big[ \sum_{i=1}^{n} \tilde{\mathbf{x}}_{ij} (1 + \log \frac{\mathbf{x}_{ij}}{\tilde{\mathbf{x}}_{ij}}) - 1 \Big].$$

Using the inequality $z \geq 1 + \log z$ and $ab \leq \frac{1}{2}(a^2 + b^2)(a \leq \frac{1}{2}(\frac{a^2}{b} + b))$, one can prove that Eq.(12) is a lower bound of Eq.(11). Thus, $Z(\mathbf{x}, \tilde{\mathbf{x}})$ is an auxiliary function of $\mathcal{L}(\mathbf{x})$. According to Eq.(9), we need to find the global maximum of $\Phi(\mathbf{x}, \tilde{\mathbf{x}})$ for $\mathbf{x}$. The gradient is

$$\frac{\partial \Phi(\mathbf{x}, \tilde{\mathbf{x}})}{\partial \mathbf{x}_{kl}} = 2(\mathbf{W}\tilde{\mathbf{x}})_{kl} \frac{\tilde{\mathbf{x}}_{kl}}{\mathbf{x}_{kl}} - \mathbf{\Lambda}_k^{+} \frac{\mathbf{x}_{kl}}{\tilde{\mathbf{x}}_{kl}} + \mathbf{\Lambda}_k^{-} \frac{\tilde{\mathbf{x}}_{kl}}{\mathbf{x}_{kl}} - \mathbf{\Gamma}_l^{+} \frac{\mathbf{x}_{kl}}{\tilde{\mathbf{x}}_{kl}} + \mathbf{\Gamma}_l^{-} \frac{\tilde{\mathbf{x}}_{kl}}{\mathbf{x}_{kl}}$$

Note that, for graph matching problem, we have $\mathbf{W}^{\mathrm{T}} = \mathbf{W}$. Thus, the second derivative is

$$\frac{\partial^2 \Phi(\mathbf{x}, \tilde{\mathbf{x}})}{\partial \mathbf{x}_{kl} \partial \mathbf{x}_{ij}} = - \Big[ \big( 2(\mathbf{W}\tilde{\mathbf{x}})_{kl} + \mathbf{\Lambda}_k^{-} + \mathbf{\Gamma}_l^{-} \big) \frac{\tilde{\mathbf{x}}_{kl}}{\mathbf{x}_{kl}^2} + \frac{1}{\tilde{\mathbf{x}}_{kl}} (\mathbf{\Lambda}_k^{+} + \mathbf{\Gamma}_l^{+}) \Big] \delta_{ki} \delta_{lj} \leq 0, \tag{13}$$

Therefore, $\Phi(\mathbf{x}, \tilde{\mathbf{x}})$ is a concave function in $\mathbf{x}$ and has a unique global maximum. It can be obtained by setting the first derivative to zero ($\frac{\partial \Phi(\mathbf{x}, \tilde{\mathbf{x}})}{\partial \mathbf{x}_{kl}} = 0$), which gives

$$\mathbf{x}_{kl} = \tilde{\mathbf{x}}_{kl} \Big[ \frac{2(\mathbf{W}\tilde{\mathbf{x}})_{kl} + \mathbf{\Lambda}_k^{-} + \mathbf{\Gamma}_l^{-}}{\mathbf{\Lambda}_k^{+} + \mathbf{\Gamma}_l^{+}} \Big]^{1/2}. \tag{14}$$

Therefore, we obtain the update rule in Eq.(5) by setting $\mathbf{x}^{(t+1)} = \mathbf{x}$ and $\mathbf{x}^{(t)} = \tilde{\mathbf{x}}$. $\square$

**Theorem 2**. *Under update Eq.(5), the converged solution $\mathbf{x}^*$ is Karush-Kuhn-Tucker (KKT) optimal.*

**Proof**. The standard Lagrangian function is

$$\mathcal{L}(\mathbf{x}) = \mathbf{x}^{\mathrm{T}} \mathbf{W} \mathbf{x} - \sum_{i=1}^{n} \mathbf{\Lambda}_i (\sum_{j=1}^{n} \mathbf{x}_{ij} - 1) - \sum_{j=1}^{n} \mathbf{\Gamma}_j (\sum_{i=1}^{n} \mathbf{x}_{ij} - 1) - \sum_{i=1}^{n} \sum_{j=1}^{n} \mathbf{\Delta}_{ij} \mathbf{x}_{ij} \tag{15}$$

Here, we use the Lagrangian function to induce KKT optimal condition. Using Eq.(15), we have

$$\frac{\partial \mathcal{L}(\mathbf{x})}{\partial \mathbf{x}_{kl}} = 2(\mathbf{W}\mathbf{x})_{kl} - \boldsymbol{\lambda}_k - \boldsymbol{\mu}_l. \tag{16}$$

The corresponding KKT condition is

$$\frac{\partial \mathcal{L}(\mathbf{x})}{\partial \mathbf{x}_{kl}} = 2(\mathbf{W}\mathbf{x})_{kl} - \mathbf{\Lambda}_k - \mathbf{\Gamma}_l - \mathbf{\Delta}_{kl} = 0 \tag{17}$$

$$\frac{\partial \mathcal{L}(\mathbf{x})}{\partial \mathbf{\Lambda}_k} = -(\sum_l \mathbf{x}_{kl} - 1) = 0 \tag{18}$$

$$\frac{\partial \mathcal{L}(\mathbf{x})}{\partial \mathbf{\Gamma}_l} = -(\sum_k \mathbf{x}_{kl} - 1) = 0 \tag{19}$$

$$\mathbf{\Delta}_{kl} \mathbf{x}_{kl} = 0. \tag{20}$$

This leads to the following KKT complementary slackness condition,

$$\big[2(\mathbf{Wx})_{kl} - \mathbf{\Lambda}_k - \mathbf{\Gamma}_l\big]\mathbf{x}_{kl} = 0. \tag{21}$$

Because $\sum_l \mathbf{x}_{kl} = 1, \sum_k \mathbf{x}_{kl} = 1$, summing over indexes $k$ and $l$ respectively, we obtain the following two group equations,

$$2\sum_{l=1}^{n} \mathbf{x}_{kl}(\mathbf{Wx})_{kl} - \sum_{l=1}^{n} \mathbf{\Gamma}_l \mathbf{x}_{kl} - \mathbf{\Lambda}_k = 0, \tag{22}$$

$$2\sum_{k=1}^{n} \mathbf{x}_{kl}(\mathbf{Wx})_{kl} - \sum_{k=1}^{n} \mathbf{\Lambda}_k \mathbf{x}_{kl} - \mathbf{\Gamma}_l = 0. \tag{23}$$

Eqs.(22, 23) can be equivalently reformulated as the following matrix forms,

$$2\operatorname{diag}(\mathbf{KX}^{\mathrm{T}}) - \mathbf{\Lambda} - \mathbf{X\Gamma} = \mathbf{0}, \tag{24}$$

$$2\operatorname{diag}(\mathbf{K}^{\mathrm{T}}\mathbf{X}) - \mathbf{\Gamma} - \mathbf{X}^{\mathrm{T}}\mathbf{\Lambda} = \mathbf{0}. \tag{25}$$

where $k = 1, 2, \cdots n$, $l = 1, 2, \cdots n$. $\mathbf{K}, \mathbf{X}$ are the matrix forms of vector $(\mathbf{Wx})$ and $\mathbf{x}$, respectively, i.e., $\mathbf{K}, \mathbf{X} \in \mathbb{R}^{n \times n}$ and $\mathbf{K}_{kl} = (\mathbf{Wx})_{kl}, \mathbf{X}_{kl} = \mathbf{x}_{kl}$. Thus, we can obtain the values for $\mathbf{\Lambda}$ and $\mathbf{\Gamma}$ as,

$$\mathbf{\Gamma} = 2(\mathbf{I} - \mathbf{X}^{\mathrm{T}}\mathbf{X})^{-1}(\operatorname{diag}(\mathbf{K}^{\mathrm{T}}\mathbf{X}) - \mathbf{X}^{\mathrm{T}}\operatorname{diag}(\mathbf{KX}^{\mathrm{T}})) \tag{26}$$

$$\mathbf{\Lambda} = 2\operatorname{diag}(\mathbf{KX}^{\mathrm{T}}) - \mathbf{X\Gamma} \tag{27}$$

On the other hand, from update Eq.(5), at convergence,

$$\mathbf{x}_{kl}^{*} = \mathbf{x}_{kl}^{*}\Big[\frac{2(\mathbf{Wx}^{*})_{kl} + \mathbf{\Lambda}_k^{-} + \mathbf{\Gamma}_l^{-}}{\mathbf{\Lambda}_k^{+} + \mathbf{\Gamma}_l^{+}}\Big]^{1/2} \tag{28}$$

Thus, we have $(2(\mathbf{Wx}^{*})_{kl} - \mathbf{\Lambda}_k - \mathbf{\Gamma}_l)\mathbf{x}_{kl}^{*2} = 0$, which is identical to the following KKT condition,

$$\big[2(\mathbf{Wx}^{*})_{kl} - \mathbf{\Lambda}_k - \mathbf{\Gamma}_l\big]\mathbf{x}_{kl}^{*} = 0. \tag{29}$$

Substituting the values of $\mathbf{\Lambda}_k, \mathbf{\Gamma}_l$ in Eq.(28) from Eqs.(26,27), we obtain update rule Eq.(5). $\square$

**Remark.** Similar to the above analysis, we can also derive another similar update as,

$$\mathbf{x}_{kl}^{(t+1)} = \mathbf{x}_{kl}^{(t)} \frac{2(\mathbf{Wx}^{(t)})_{kl} + \mathbf{\Lambda}_k^{-} + \mathbf{\Gamma}_l^{-}}{\mathbf{\Lambda}_k^{+} + \mathbf{\Gamma}_l^{+}}. \tag{30}$$

The optimality and convergence of this update are also guaranteed. We omit the further discussion of them due to the lack of space. In real application, one can use both of these two update algorithms (Eq.(5), Eq.(30)) to obtain better results.

## 5 Sparsity and Discrete Solution

One property of the proposed MPGM is that it can result in a sparse optimal solution, although the discrete binary constraint have been dropped in MPGM optimization process. This suggests that MPGM can search for an optimal solution nearly on the permutation domain $\mathcal{P}$, i.e., the boundary of the doubly stochastic domain $\mathcal{D}$. Unfortunately, here we cannot provide a theoretical proof on the sparsity of MPGM solution, but demonstrate it experimentally.

Figure 1 (a) shows the solution $\mathbf{x}^{(t)}$ across different iterations. Note that, regardless of initialization, as the iteration increases, the solution vector $\mathbf{x}^{(t)}$ of MPGM becomes more and more sparse and converges to a discrete binary solution. Note that, in MPGM update Eq.(5), when $\mathbf{x}_{kl}^t$ closes to zero, it can keep closing to zero in the following update process because of the particular multiplicative operation. Therefore, as the iteration increases, the solution vector $\mathbf{x}^{t+1}$ is guaranteed to be more sparse than solution vector $\mathbf{x}^t$. Figure 1 (b) shows the objective and sparsity[2] of the solution vector $\mathbf{x}^{(t)}$. We can observe that (1) the objective of $\mathbf{x}^{(t)}$ increases and converges after some iterations, demonstrating the convergence of MPGM algorithm. (2) The sparsity of the solution $\mathbf{x}^{(t)}$ increases and converges to the baseline, which demonstrates the ability of MPGM algorithm to maintain the discrete constraint in the converged solution.

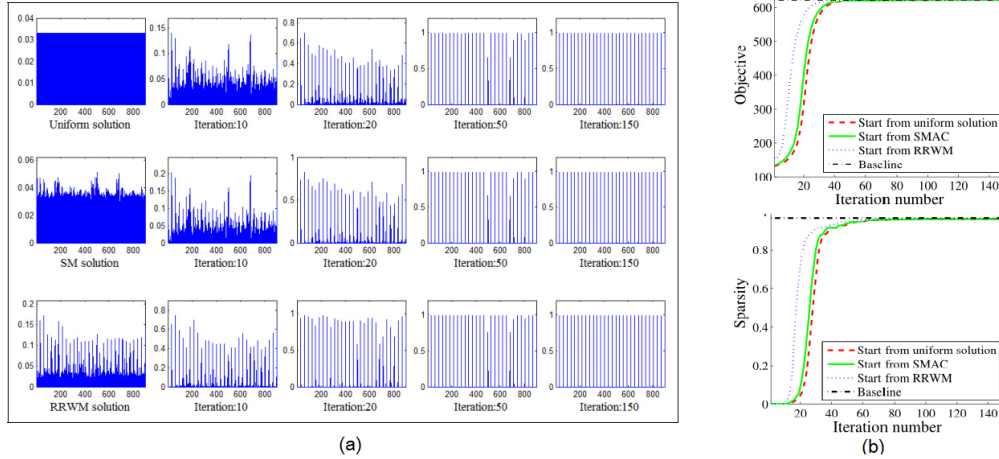

Figure 1: (a) Solution vector $\mathbf{x}^{(t)}$ of MPGM across different iterations (top: start from uniform solution; middle: start from SM solution; bottom: start from RRWM solution).

## 6 Experiments

We have applied MPGM algorithm to several matching tasks. Our method has been compared with some other state-of-the-art methods including SM [15], IPFP [16], SMAC [5], RRWM [3] and FGM [24]. We implemented IPFP [16] with two versions: (1) IPFP-U that is initialized by the uniform solution; (2) IPFP-S that is initialized by SM method [15]. In experiments, we initialize our MPGM with uniform solution and obtain similar results when initializing with SM solution.

### 6.1 Synthetic Data

Similar to the works [3, 24], we have randomly generated data sets of $n_{in}$ 2D points as inlier nodes for $G$. We obtain the corresponding nodes in graph $G'$ by transforming the whole point set with a random rotation and translation and then adding Gaussian noise $N(0, \sigma)$ to the point positions from graph $G$. In addition, we also added $n_{out}$ outlier nodes in both graphs respectively at random positions. The affinity matrix $\mathbf{W}$ has been computed as $\mathbf{W}_{ij,kl} = \exp(-\|\mathbf{r}_{ik} - \mathbf{r}'_{jl}\|_F^2/0.0015)$, where $\mathbf{r}_{ik}$ is the Euclidean distance between two nodes in $G$ and similarly for $\mathbf{r}'_{jl}$.

Figure 2 summarizes the comparison results. We can note that: (1) similar to IPFP [16] and FGM [24] which return discrete matching solutions, MPGM always generates sparse solutions on doubly stochastic domain. (2) MPGM returns higher objective score and accuracy than IPFP [16] and FGM [24] methods, which demonstrate that MPGM can find the sparse solution more optimal than these methods. (3) MPGM generally performs better than the continuous domain methods including SM [15], SMAC [5] and RRWM [3]. Comparing with these methods, MPGM incorporates the doubly stochastic constraint more naturally and thus finds the solution more optimal than RRWM method. (4) MPGM generally has similar time cost with RRWM [3]. We have not shown the time cost of FGM [24] method in Fig.2, because FGM uses a hybrid optimization method and has obviously higher time cost than other methods.

### 6.2 Image Sequence Data

In this section, we perform feature matching on CMU and YORK house sequences [3, 2, 18]. For CMU "hotel" sequence, we have matched all images spaced by 5, 10 $\cdots$ 75 and 80 frames and computed the average performances per separation gap. For YORK house sequence, we have matched all images spaced by 1, 2 $\cdots$ 8 and 9 frames and computed the average performances per separation gap. The affinity matrix has been computed by $\mathbf{W}_{ij,kl} = \exp(-\|\mathbf{r}_{ik} - \mathbf{r}'_{jl}\|_F^2/1000)$, where $\mathbf{r}_{ik}$ is the Euclidean distance between two points.

Figure 3 summarizes the performance results. It is noted that MPGM outperforms the other methods in both objective score and matching accuracy, indicating the effectiveness of MPGM method. Also,

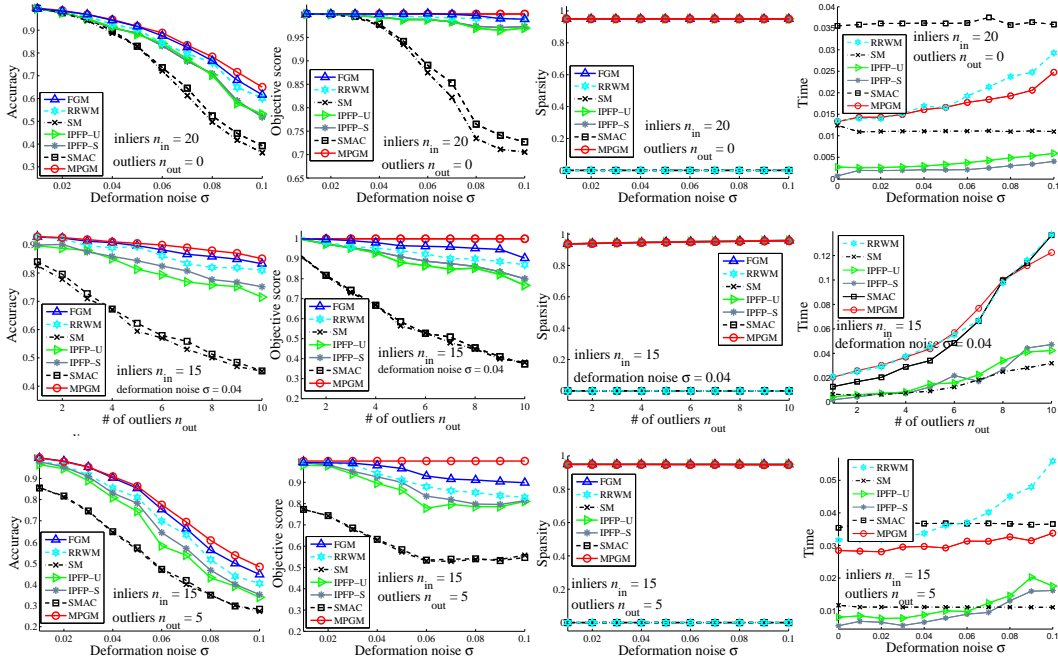

Figure 2: Comparison results of different methods on synthetic point sets matching

MPGM can generate sparse solutions. These are generally consistent with the results on the synthetic data experiments and further demonstrate the benefits of MPGM algorithm.

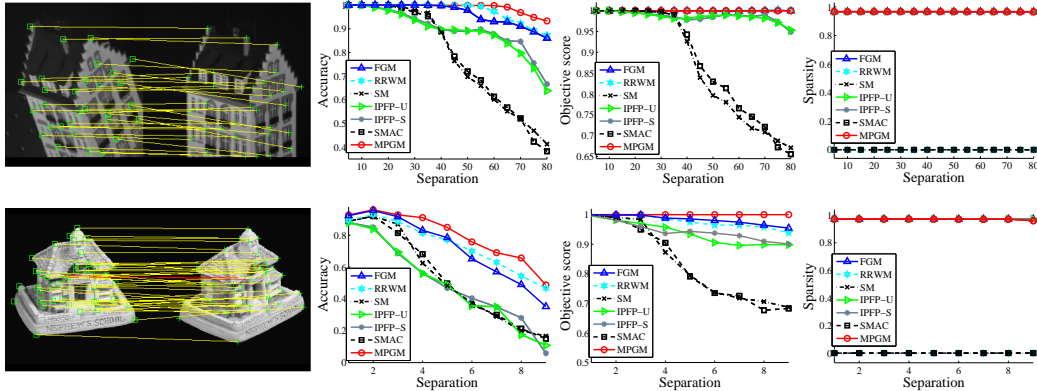

Figure 3: Comparison results of different methods on CMU and YORK image sequences. Top: CMU images; Bottom: YORK images.

## 6.3 Real-world Image Data

In this section, we tested our method on some real-world image datasets. We evaluate our MPGM on the dataset [17] whose images are selected from Pascal 2007 [3]. In this dataset, there are 30 pairs of car images and 20 pairs of motorbike images. For each image pair, feature points and ground-truth matches were manually marked and each pair contains 30-60 ground-truth correspondences.

The affinity between two nodes is computed as $\mathbf{W}_{ij,ij} = \exp(\frac{-|p_i - p'_j|}{0.05})$, where $p_i$ is the orientation of normal vector at the sampled point (node) $i$ to the contour, similarly to $p'_j$. Also, the affinity

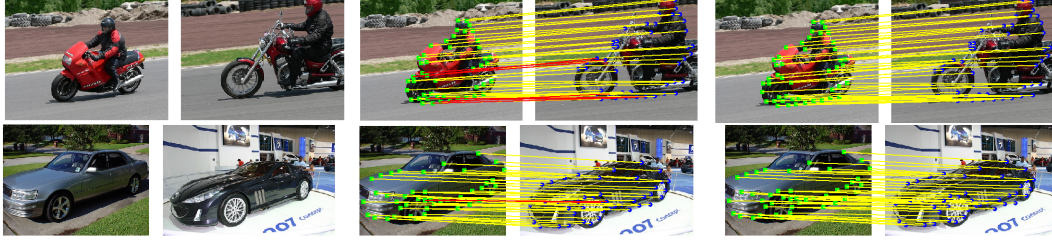

Figure 4: Some examples of image matching on Pascal 2007 dataset (LEFT: original image pair, MIDDLE: FGM result, RIGHT: MPGM result. Incorrect matches are marked by red lines)

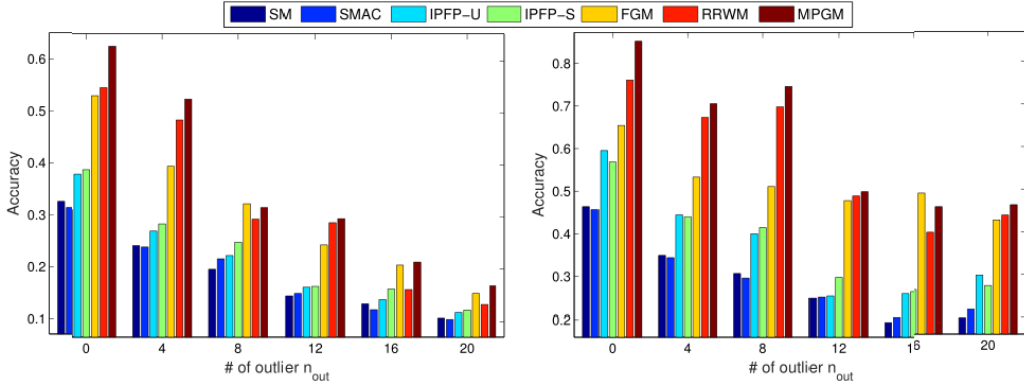

Figure 5: Comparison results of different graph matching methods on the Pascal 2007 dataset

between two correspondences has been computed as $\mathbf{W}_{ij,kl} = \exp(\frac{-|d_{ik} - d'_{jl}|}{0.15})$, where $d_{ik}$ denotes the Euclidean distance between feature point $i$ and $k$, similarly to $d'_{jl}$. Some matching examples are shown in Figure 4. To test the performance against outlier noise, we have randomly added 0-20 outlier features for each image pair. The overall results of matching accuracy across different outlier features are summarized in Figure 5. From Figure 5, we can note that MPGM outperforms the other competing methods including RRWM [3] and FGM [24], which further demonstrates the effectiveness and practicality of MPGM on conducting real-world image matching tasks.

## 7 Conclusions and Future work

This paper presents an effective algorithm, Multiplicative Update Graph Matching (MPGM), that develops a multiplicative update technique to solve the QP matching problem with doubly stochastic mapping constraint. The KKT optimality and convergence properties of MPGM algorithms are theoretically guaranteed. We show experimentally that MPGM solution is sparse and thus approximately incorporates the discrete constraint in optimization naturally. In our future, the theoretical analysis on the sparsity of MPGM needs to be further studied. Also, we will incorporate our MPGM in some path-following strategy to find a more optimal solution for the matching problem. We will adapt the proposed algorithm to solve some other optimization problems with doubly stochastic constraint in machine learning and computer vision area.

## Acknowledgment

This work is supported by the NBRPC 973 Program (2015CB351705); National Natural Science Foundation of China (61602001,61671018, 61572030); Natural Science Foundation of Anhui Province (1708085QF139); Natural Science Foundation of Anhui Higher Education Institutions of China (KJ2016A020); Co-Innovation Center for Information Supply & Assurance Technology, Anhui University; The Open Projects Program of National Laboratory of Pattern Recognition.

## Footnotes

[1]Here, we focus on equal-size graph matching problem. For graphs with different sizes, one can add dummy isolated nodes into the smaller graph and transform them to equal-size case [21, 10]

[2]Sparsity measures the percentage of zero (close-to-zero) elements in $\mathbf{Z}$. Firstly, set the threshold $\epsilon = 0.001 \times \operatorname{mean}(\mathbf{Z})$, then renew $\mathbf{Z}_{ij} = 0$ if $\mathbf{Z}_{ij} \leq \epsilon$. Finally, the sparsity is defined as the percentage of zero elements in the renewed $\mathbf{Z}$.

[3]http://www.pascalnetwork.org/challenges/VOC/voc2007/workshop/index.html

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
