[Reviews · NeurIPS 2017]

Reviewer 1



The paper presents a new method called Multiplicative Update Graph Matching (MPGM) to provides a multiplicative update technique for the general Quadratic Programming problem. Compared to existing methods, the update steps have a closed form solution, the convergence is guaranteed, and the doubly stochastic constraints are directly integrated in the optimization process. The paper is technically sound, and the claims seem theoretically and experimentally well supported. The paper writing is clear and the organization is neat. Without knowing too much about graph matching and its related literature, the approach seems novel, and given the description and the qualitative and quantitative results, the advance over state-of-the-art seems sufficient. Graph Matching is a commonly known problem. Having an algorithm with guaranteed convergence

Reviewer 2



This paper presents a novel method (MPMG) to solve QP graph matching problem. The graph matching problem is formulated as argmax_x(x^TWx) where W encodes the pair-to-pair and node-to-node affinities and x is the desired permutation matrix (in vector form). The desired solution is a permutation encoding the optimal matching, also expressed as doubly stochastic (entries >= 0, rows/columns sum to 1) and discrete (entries in {0,1}). The standard approaches to solving the QP is to relax the NP-hard problem (relaxation can be for either the doubly-stochastic constraint or the discrete constraint). This proposed MPMG solution follows the Lagrange multiplier technique which moves the doubly-stochastic constraint into the error term. This proposed approach, along with proofs for convergence and KKT-optimality are a a novel contribution for this graph matching formulation. Experimentally, evidence is provided that shows the approach converges near a sparse/discrete solution even though the constraint is not explicitly modeled in the solution, which is the most promising feature of the method. However, there are some concerns regarding the experimental details which might jeopardize the strength of the empirical claims. Other state of the art techniques do rely on iterative solutions, so the paper should mention the convergence criterion especially for experiments like figure 1 where the MPMG is initialized with RRWM, it would be useful to make sure that the convergence criterion for RRWM is appropriately selected for these problems. Following up on the previous comment, it seems initialization with RRWM is used for FIgure 1 but initialization with SM for the subsequent experiments. Furthermore, regarding the experiments, how is performance time computed? Is this including time for the initialization procedure, or just the subsequent MPGM iterations after initialization? Also, RRWM also requires an initialization. In [3] the basic outline of the algorithm uses uniform, so is it a fair comparison at all for MPMG that uses a established algorithm as initialization vs RRWM which uses uniform? If the details have been handled properly perhaps more explanation should be given in the text to clarify these points. In light of these points it is hard to take anything meaningful away from the experiments provided. In the experiments, how are the node-to-node affinities determined (diagonals of W)? I think this is not described in the paper. Image feature point matching would not actually be solved as a graph matching problem in any practical scenario (this is true for a number of reasons not limited to number of features, pair-to-pair affinities, robustness to large # of outliers). As the paper suggests, there are many other practical labeling problems which are typically solved using this QP formulation and perhaps the paper would have more value stronger

Reviewer 3



The purpose of the paper is to find a way to iteratively update a solution that will converge to the KKT optimum solution. Starting with the KKT solution (line 141), the authors find a way to write it in a iterative recurrence (by splitting the square term into x(t) * x(t+1)). Then, they construct an auxiliary lower bound function (in 2 variables: the old and the new solution) for the Lagrangian such that its solution gives the update rule. The lower bound is a concave function which, except for the variable in the Lagrangian, brings in a new variable (with the role of the old solution), having as intention the recurrence rule between new and old solution. Moreover, the authors force it by construction to converge to a sparse solution (empirically proven only). This part should be analyzed more. Did you construct more update rules, but with non-sparse results? You should find the particularities in your update rule that make the solution sparse.